# Long Colimits of Topological Groups III: Homeomorphisms of Products and Coproducts

Rafael Dahmen [1] and Gábor Lukács [2,*]

1 Department of Mathematics, Karlsruhe Institute of Technology, D-76128 Karlsruhe, Germany; rafael.dahmen@kit.edu
2 Department of Mathematics and Statistics, Dalhousie University, Halifax, NS B3H 3J5, Canada
* Correspondence: lukacs@topgroups.ca

**Abstract:** The group of compactly supported homeomorphisms on a Tychonoff space can be topologized in a number of ways, including as a colimit of homeomorphism groups with a given compact support or as a subgroup of the homeomorphism group of its Stone-Čech compactification. A space is said to have the *Compactly Supported Homeomorphism Property* (*CSHP*) if these two topologies coincide. The authors provide necessary and sufficient conditions for finite products of ordinals equipped with the order topology to have CSHP. In addition, necessary conditions are presented for finite products and coproducts of spaces to have CSHP.

**Keywords:** long colimit; topological group; homeomorphism group

## 1. Introduction

Given a compact space $K$, it is well known that the homeomorphism group $\mathrm{Homeo}(K)$ is a topological group with the compact-open topology [1]. If $X$ is assumed to be only Tychonoff, then for every compact subset $K \subseteq X$, the group $\mathrm{Homeo}_K(X)$ of homeomorphisms supported in $K$ (i.e., identity on $X \backslash K$) is a topological group with the compact-open topology; however, the full homeomorphism group $\mathrm{Homeo}(X)$ equipped with the compact-open topology need not be a topological group [2]. Nevertheless, $\mathrm{Homeo}(X)$ can be turned into a topological group by embedding it into $\mathrm{Homeo}(\beta X)$, the homeomorphism group of the Stone-Čech compactification of $X$. The latter topology has also been studied under the name of *zero-cozero topology* [3,4].

For a Tychonoff space $X$, let $\mathscr{K}(X)$ denote the family of compact subsets of $X$. In light of the foregoing, the group $\mathrm{Homeo}_{cpt}(X) := \bigcup_{K \in \mathscr{K}(X)} \mathrm{Homeo}_K(X)$ of the compactly supported homeomorphisms of $X$ admits three seemingly different topologies, listed from the finest to the coarsest:

(a) the finest topology making all inclusions $\mathrm{Homeo}_K(X) \longrightarrow \mathrm{Homeo}_{cpt}(X)$ continuous (i.e., the colimit in the category of topological spaces and continuous functions);

(b) the finest *group* topology making all inclusions $\mathrm{Homeo}_K(X) \longrightarrow \mathrm{Homeo}_{cpt}(X)$ continuous (i.e., the colimit in the category of topological *groups* and continuous *homomorphisms*); and

(c) the topology induced by $\mathrm{Homeo}(\beta X)$.

Recall that the groups $\{\mathrm{Homeo}_K(X)\}_{K \in \mathscr{K}(X)}$ are said to have the *Algebraic Colimit Property* (*ACP*) if the first and the second topologies coincide [5,6]. Recall further that a space $X$ is said to have the *Compactly Supported Homeomorphism Property* (*CSHP*) if the first and the last topologies coincide [5]), in which case all three topologies are equal.

In a previous work, the authors gave sufficient conditions for a finite product of ordinals to have CSHP ([5], Theorem D(c)).The main result of this paper is that the same

conditions are also necessary, thereby providing a complete characterization of CSHP among such spaces.

**Theorem A.** *Let $X = \lambda_1 \times \cdots \lambda_k \times \mu_1 \times \cdots \times \mu_l$ equipped with the product topology, where $\lambda_1, \ldots, \lambda_k$ are infinite limit ordinals and $\mu_1, \ldots, \mu_k$ are successor ordinals. The space X has CSHP if and only if there is an uncountable regular cardinal $\kappa$ such that $\lambda_1 = \cdots = \lambda_k = \kappa$ and $\mu_i \leq \kappa$ for every $i = 1, \ldots, l$.*

**Example 1.** *By Theorem A, the spaces $\omega_1 \times \omega_2$ and $\omega_1 \times (\omega_1 + 1)$ (with the product topology) and $\omega_2 + \omega_1$ (sum of ordinals with the order topology) do not have CSHP. Furthermore, the disjoint union (coproduct) $\omega_1 \amalg \omega_2$ does not have CSHP either (see Corollary 2).*

The proof of Theorem A is based on results of general applicability about CSHP of products and coproducts of spaces. For an infinite cardinal $\tau$, a subset $S$ of a space $X$ is said to be *$\tau$-discrete in $X$* if every subset of $S$ of cardinality less than $\tau$ is closed in $X$. If $S$ is $\tau$-discrete in $X$, then every subset of $S$ of cardinality less than $\tau$ is discrete. Being $\tau$-discrete in $X$ is equivalent to being closed and discrete in a certain finer topology (Proposition 1). Recall that the *cofinality* $\mathrm{cf}(\mathbb{I}, \leq)$ of a partially ordered set $(\mathbb{I}, \leq)$ is the smallest cardinal of a cofinal set contained in $\mathbb{I}$.

**Theorem B.** *Let $Y$ be a compact Hausdorff space, $Z$ be a zero-dimensional locally compact Hausdorff pseudocompact space that is not compact, and $\tau := \mathrm{cf}(\mathscr{K}(Z), \subseteq)$. If $\mathrm{Homeo}(Y)$ contains a $\tau$-discrete subset of cardinality $\tau$ that is not closed, then the product $Y \times Z$ does not have CSHP.*

Recall that the *support* of a homeomorphism $h$ of a space $X$ is

$$\mathrm{supp}\, h := \mathrm{cl}_X\{x \in X \mid h(x) \neq x\}.$$

**Theorem C.** *Let $Y$ be a compact Hausdorff space, $Z$ a locally compact Hausdorff space, and $\{K_\alpha\}_{\alpha < \tau}$ a cofinal family in $\mathscr{K}(Z)$, where $\tau$ is an infinite cardinal. Suppose further that*

(I)　　　$\mathrm{Homeo}(Y)$ *contains a $\tau$-discrete subset of cardinality $\tau$ that is not closed; and*
(II)　　$\mathrm{Homeo}_{cpt}(Z)$ *contains a net $(g_\beta)_{\beta < \tau}$ of distinct elements such that $\lim g_\beta = \mathrm{id}_Z$ and $\mathrm{supp}\, g_\beta \not\subseteq K_\alpha$ whenever $\alpha < \beta$.*

*Then the coproduct (disjoint union) $Y \amalg Z$ does not have CSHP.*

In order to invoke Theorems B and C, one needs to ensure that $\mathrm{Homeo}(Y)$ contains a $\tau$-discrete subset of cardinality $\tau$ that is not closed. For spaces that are of interest to us in this paper, this is guaranteed by the next theorem.

**Theorem D.** *Let $\alpha$ be an infinite limit ordinal with $\tau := \mathrm{cf}(\alpha)$, and put $Y = \alpha + 1$ with the order topology. Then $\mathrm{Homeo}(Y)$ contains a $\tau$-discrete subset of cardinality $\tau$ that is not closed.*

The paper is structured as follows. In Section 2, we provide some preliminary results that are used throughout the paper. In Section 3, we prove Theorems B and C, while the proof of Theorem D is presented in Section 4. Lastly, Theorem A is proven in Section 5.

## 2. Preliminaries

Let $\tau$ be an infinite cardinal. For a topological space $(X, \mathcal{T})$, the subsets of $X$ of cardinality less than $\tau$ form a directed system with respect to inclusion. We put

$$(X, \mathcal{T}_{<\tau}) := \mathrm{colim}\{Y \mid Y \subseteq X, |Y| < \tau\},$$

where the colimit is formed in the category Top of topological spaces and their continuous maps.

**Proposition 1.** *Let $\tau$ be an infinite cardinal and $(X, \mathcal{T})$ a topological space. A subset $S \subseteq X$ is $\tau$-discrete in $X$ if and only if $S$ is closed and discrete in $(X, \mathcal{T}_{<\tau})$.*

**Proof.** Suppose that $S \subseteq X$ is $\tau$-discrete. Then $|S \cap Y| < \tau$ for every $Y \subseteq X$ with $|Y| < \tau$, and thus $S \cap Y$ is closed in $X$; in particular, $S \cap Y$ is closed $Y$. Therefore, $S$ is closed in $(X, \mathcal{T}_{<\tau})$. Let $s_0 \in S$. Then $S \backslash \{s_0\}$ is also $\tau$-discrete, and consequently, by the previous argument, closed in $(X, \mathcal{T}_{<\tau})$. Hence, the singleton $\{s_0\}$ is open in $S$ in the topology induced by $(X, \mathcal{T}_{<\tau})$. This shows that $S$ is discrete in $(X, \mathcal{T}_{<\tau})$.

Conversely, suppose that $S \subseteq X$ is closed and discrete in $(X, \mathcal{T}_{<\tau})$. Let $A \subseteq S$ be such that $|A| < \tau$. We show that $A$ is closed in $(X, \mathcal{T})$. Let $y_0 \in X \backslash A$, and put $Y := A \cup \{y_0\}$. Then $|Y| < \tau$, and so $S \cap Y$ is closed and discrete in $Y$. If $y_0 \in S$, then $S \cap Y = Y$ is discrete, and so $A = Y \backslash \{y_0\}$ is closed in $Y$. If $y_0 \notin S$, then $S \cap Y = A$ is closed in $Y$. In both cases, $y_0 \notin \mathrm{cl}_Y A$, and therefore $y_0 \notin \mathrm{cl}_X A$. This shows that $A$ is closed in $X$, as desired.  $\square$

**Proposition 2.** *Let $\tau$ be an infinite cardinal, $f \colon (X, \mathcal{T}) \to (Y, \mathcal{T}')$ be a continuous map between Hausdorff spaces, and $S$ a subset of $X$ such that $f_{|S}$ is injective. If $f(S)$ is $\tau$-discrete in $Y$, then $S$ is $\tau$-discrete in $X$.*

**Proof.** By Proposition 1, it suffices to show that $S$ is closed and discrete in $(S, \mathcal{T}_{<\tau})$. Put $S' := f(S)$. Since the $< \tau$-topology is functorial, $f \colon (X, \mathcal{T}_{<\tau}) \to (Y, \mathcal{T}'_{<\tau})$ is continuous, and in particular, $f_{|S} \colon (S, \mathcal{T}_{<\tau}) \to (S', \mathcal{T}'_{<\tau})$ is continuous and bijective. By Proposition 1, $(S', \mathcal{T}'_{<\tau})$ is discrete and $S'$ is closed in $(Y, \mathcal{T}'_{<\tau})$. Thus, $(S, \mathcal{T}_{<\tau})$ is discrete, and furthermore

$$f(\mathrm{cl}_{(X, \mathcal{T}_{<\tau})} S) \subseteq \mathrm{cl}_{(Y, \mathcal{T}'_{<\tau})} S' = S'. \tag{1}$$

To show that $S$ is closed in $(X, \mathcal{T}_{<\tau})$, let $s_0 \in \mathrm{cl}_{(X, \mathcal{T}_{<\tau})} S$. Then there is a net $(s_\alpha) \subseteq S$ such that $s_\alpha \xrightarrow{(X, \mathcal{T}_{<\tau})} s_0$, and so $f(s_\alpha) \xrightarrow{(Y, \mathcal{T}'_{<\tau})} f(s_0)$. By (1), $f(s_0) \in S'$. Since $S'$ is discrete in $(Y, \mathcal{T}'_{<\tau})$, the net $(f(s_\alpha))$ is eventually constant. Therefore, $(s_\alpha)$ is eventually constant, because $f_{|S}$ is injective. Hence, $s_0 \in S$, because $(X, \mathcal{T})$ is Hausdorff, and in particular, $(X, \mathcal{T}_{<\tau})$ is Hausdorff.  $\square$

The next lemma allows one to show that a space does not have CSHP by constructing a suitable $\tau$-discrete set in its homeomorphism group.

**Lemma 1.** *Let $X$ be a topological space and $\{X_\alpha\}_{\alpha \in \mathbb{I}}$ a directed system of subsets of $X$ such that $X = \bigcup_{\alpha \in \mathbb{I}} X_\alpha$. Suppose that there is an infinite cardinal $\tau$ and a subset $S \subseteq X$ such that:*

(1)     *$S$ is $\tau$-discrete in $X$;*
(2)     *$|S \cap X_\alpha| < \tau$ for every $\alpha \in \mathbb{I}$; and*
(3)     *$S$ is not closed in $X$.*

*Then $X \neq \underset{\alpha \in \mathbb{I}}{\mathrm{colim}}\, X_\alpha$.*

**Proof.** Let $S \subseteq X$ be a subset with properties (1)–(3). By (1) and (2), $S \cap X_\alpha$ is closed in $X$ for every $\alpha \in \mathbb{I}$; in particular, $S \cap X_\alpha$ is closed in $X_\alpha$ for every $\alpha \in \mathbb{I}$. Thus, $S$ is closed in $\underset{\alpha \in \mathbb{I}}{\mathrm{colim}}\, X_\alpha$. By (3), $S$ is not closed in $X$. Therefore, the two topologies are distinct.  $\square$

Lastly, recall that CSHP is inherited by clopen subsets.

**Lemma 2** ([5], 5.3(b) and 5.6)**.** *Let $X$ be a Tychonoff space.*

(a)     *If $A \subseteq X$ is a clopen subset and $X$ has CSHP, then so does $A$.*
(b)     *If $X$ contains an infinite discrete clopen subset, then $X$ does not have CSHP.*

## 3. Products and Coproducts with Compact Spaces

In this section, we prove Theorems B and C. Before we prove Theorem B, we need a technical proposition about the existence of cofinal subsets with small down-sets.

**Proposition 3.** *Let $(\mathbb{I}, \leq)$ be a poset and put $\tau := \mathrm{cf}(\mathbb{I}, \leq)$. Then every cofinal subset of $\mathbb{I}$ contains a cofinal subset $J$ of cardinality $\tau$ such that $|\{b \in J \mid b \leq a\}| < \tau$ for every $a \in J$.*

**Proof.** Let $C \subseteq \mathbb{I}$ be a cofinal subset. Without loss of generality, we may assume that $|C| = \tau$. Let $C = \{c_\alpha \mid \alpha < \tau\}$ be an enumeration of $C$. We define $\{\alpha_\gamma\}_{\gamma < \tau}$ inductively as follows. We put $\alpha_0 := 0$. For $0 < \gamma < \tau$, suppose that $\alpha_\delta$ has already been defined for all $\delta < \gamma$. We observe that $\{c_{\alpha_\beta}\}_{\beta < \gamma}$ is not cofinal in $\mathbb{I}$, because its cardinality is smaller than $\tau$. Thus, $\{\alpha < \tau \mid (\forall \beta < \gamma)(c_\alpha \nleq c_{\alpha_\beta})\}$ is non-empty. Put

$$\alpha_\gamma := \min\{\alpha < \tau \mid (\forall \beta < \gamma)(c_\alpha \nleq c_{\alpha_\beta})\}.$$

Put $J := \{c_{\alpha_\gamma} \mid \gamma < \tau\}$. It follows from the construction of $\{\alpha_\gamma\}_{\gamma < \tau}$ that

$$c_{\alpha_\gamma} \nleq c_{\alpha_\beta} \text{ for every } \beta < \gamma < \tau. \tag{2}$$

In other words, if $c_{\alpha_\gamma} \leq c_{\alpha_\beta}$, then $\gamma \leq \beta$. Therefore, $|\{b \in J \mid b \leq a\}| < \tau$ for every $a \in J$.

It remains to show that $J$ is cofinal in $\mathbb{I}$. To that end, let $x \in \mathbb{I}$. Since $C$ is cofinal in $\mathbb{I}$, the set $\{\beta < \tau \mid x \leq c_\beta\}$ is non-empty. Put $\delta := \min\{\beta < \tau \mid x \leq c_\beta\}$. It follows from the construction of $\delta$ that

$$c_\delta \nleq c_\varepsilon \text{ for every } \varepsilon < \delta. \tag{3}$$

It follows from the construction of the $\{\alpha_\gamma\}_{\gamma < \tau}$ that they are strictly increasing, and in particular, $\delta \leq \alpha_\delta$. Thus, $\{\mu < \tau \mid \delta \leq \alpha_\mu\}$ is non-empty. Put $\gamma := \min\{\mu < \tau \mid \delta \leq \alpha_\mu\}$. For every $\beta < \gamma$, one has $\alpha_\beta < \delta$, and thus, by (3), $c_\delta \nleq c_{\alpha_\beta}$. Consequently,

$$\delta \in \{\alpha < \tau \mid (\forall \beta < \gamma)(c_\alpha \nleq c_{\alpha_\beta})\}.$$

Therefore,

$$\alpha_\gamma = \min\{\alpha < \tau \mid (\forall \beta < \gamma)(c_\alpha \nleq c_{\alpha_\beta})\} \leq \delta.$$

Hence, $\alpha_\gamma = \delta$, and $x \leq c_\delta = c_{\alpha_\gamma} \in J$. $\square$

**Theorem B.** *Let $Y$ be a compact Hausdorff space, $Z$ be a zero-dimensional locally compact Hausdorff pseudocompact space that is not compact, and $\tau := \mathrm{cf}(\mathscr{K}(Z), \subseteq)$. If $\mathrm{Homeo}(Y)$ contains a $\tau$-discrete subset of cardinality $\tau$ that is not closed, then the product $Y \times Z$ does not have CSHP.*

**Proof.** Since $Y$ is compact and $Z$ is pseudocompact, the product $Y \times Z$ is also pseudocompact ([7], 3.10.27), and by Glicksberg's Theorem ([8], Theorem 1), $\beta(Y \times Z) \cong Y \times \beta Z$.

Let $\{C_\alpha\}_{\alpha < \tau}$ be a cofinal family in $(\mathscr{K}(Z), \subseteq)$. Without loss of generality, we may assume that each $C_\alpha$ is open in $Z$, and $\bigcap_{\alpha < \tau} C_\alpha \neq \varnothing$. Using Proposition 3, one may pick a cofinal subfamily $\{K_\alpha\}_{\alpha < \tau}$ of $\{C_\alpha\}_{\alpha < \tau}$ such that

$$|\{\beta \mid K_\beta \subseteq K_\alpha\}| < \tau \text{ for every } \alpha < \tau. \tag{4}$$

Since $Y$ is compact, the family $\{Y \times K_\alpha\}_{\alpha < \tau}$ is cofinal in $(\mathscr{K}(Y \times Z), \subseteq)$; in particular, it is directed.

Put $G := \mathrm{Homeo}_{cpt}(Y \times Z)$ and $G_\alpha := \mathrm{Homeo}_{Y \times K_\alpha}(Y \times Z)$. We construct a subset $S \subseteq G$ that satisfies the conditions of Lemma 1:

(1)　$S$ is $\tau$-discrete in $G$;
(2)　$|S \cap G_\alpha| < \tau$ for all $\alpha < \tau$; and
(3)　$\mathrm{id}_{Y \times Z} \in \overline{S} \setminus S$.

This will show that $G \neq \mathrm{colim}_{\alpha < \tau} G_\alpha$, and thus $Y \times Z$ does not have CSHP.

Let $S' \subseteq \mathrm{Homeo}(Y)$ be a $\tau$-discrete subset such that $|S'| = \tau$ and $S'$ is not closed. Without loss of generality, we may assume that $\mathrm{id}_Y \in \overline{S'} \backslash S'$. Let $S' = \{f_\alpha \mid \alpha < \tau\}$ be an injective enumeration of $S'$. For $\alpha < \tau$, put

$$h_\alpha \colon Y \times Z \longrightarrow Y \times Z$$

$$(y, z) \longmapsto \begin{cases} (f_\alpha(y), z) & z \in K_\alpha \\ (y, z) & z \notin K_\alpha. \end{cases} \tag{5}$$

Since $h_\alpha$ is a homeomorphism on the clopen set $Y \times K_\alpha$ and $h_\alpha$ is the identity on the clopen set $Y \times (Z \backslash K_\alpha)$, one has $h_\alpha \in G$ in for every $\alpha < \tau$.

Put $S := \{h_\alpha \mid \alpha < \tau\}$. We verify that $S$ satisfies properties (1), (2), and (3).

(1) Let $\pi_Y \colon Y \times Z \to Y$ and $\pi_Z \colon Y \times Z \to Z$ denote the respective projections and put

$$H := \{h \in G \mid \pi_Z h = \pi_Z\}. \tag{6}$$

Since $H$ is a closed subgroup of $G$, it suffices to show that $S$ is $\tau$-discrete in $H$. Fix $z_0 \in \bigcap_{\alpha < \tau} K_\alpha$, and define $\iota_0 \colon Y \to Y \times Z$ by $\iota_0(y) = (y, z_0)$. The composite

$$\mathscr{C}(Y \times \beta Z, Y \times \beta Z) \xrightarrow{\mathscr{C}(\beta\iota_0, Y \times \beta Z)} \mathscr{C}(Y, Y \times \beta Z) \xrightarrow{\mathscr{C}(Y, \pi_Y)} \mathscr{C}(Y, Y) \tag{7}$$

is continuous ([7], 3.4.2), where the function spaces are equipped with the compact-open topology. Thus, its restriction to $H$,

$$\Gamma \colon H \longrightarrow \mathrm{Homeo}(Y)$$

$$h \longmapsto \pi_Y h \iota_0 \tag{8}$$

is a continuous group homomorphism. The restriction $\Gamma_{|S}$ is injective (because $\Gamma(h_\alpha) = f_\alpha$), and $\Gamma(S) = S'$ is $\tau$-discrete in $\mathrm{Homeo}(Y)$. Therefore, by Proposition 2, $S$ is $\tau$-discrete in $H$.

(2) For $\beta < \tau$, $h_\beta \in S \cap G_\alpha$ if and only if $(\mathrm{supp}\, f_\beta) \times K_\beta = \mathrm{supp}\, h_\beta \subseteq Y \times K_\alpha$, or equivalently, $K_\beta \subseteq K_\alpha$ ($\mathrm{supp}\, f_\beta \neq \varnothing$ because $\mathrm{id}_Y \notin S$). Therefore, by (4),

$$|S \cap G_\alpha| = |\{\beta \mid K_\beta \subseteq K_\alpha\}| < \tau. \tag{9}$$

(3) Since $f_\alpha \neq \mathrm{id}_Y$ for every $\alpha < \tau$, it follows that $h_\alpha \neq \mathrm{id}_{Y \times Z}$, and thus $\mathrm{id}_{Y \times Z} \notin S$. It remains to show that $\mathrm{id}_{Y \times Z} \in \bar{S}$. To that end, let $W$ be an entourage of the diagonal in $(Y \times \beta Z)^2$. Then

$$\{(u_1, v_1, u_2, v_2) \mid (u_1, u_2) \in U, (v_1, v_2) \in V\} \subseteq W \tag{10}$$

for some entourage $U$ of the diagonal in $Y \times Y$ and entourage $V$ of the diagonal in $\beta Z \times \beta Z$ ([7], 8.2.1). Since $\mathrm{id}_Y \in \overline{S'}$, there is $\gamma < \tau$ such that $(y, f_\gamma(y)) \in U$ for every $y \in Y$. Therefore, $(y, z, \beta h_\gamma(y, z)) \in W$ for every $(y, z) \in Y \times \beta Z$. Hence, $\mathrm{id}_{Y \times Z} \in \bar{S}$. □

**Theorem C.** *Let $Y$ be a compact Hausdorff space, $Z$ a locally compact Hausdorff space, and $\{K_\alpha\}_{\alpha < \tau}$ a cofinal family in $\mathscr{K}(Z)$, where $\tau$ is an infinite cardinal. Suppose further that:*

(I)　　　*$\mathrm{Homeo}(Y)$ contains a $\tau$-discrete subset of cardinality $\tau$ that is not closed; and*

(II)　　*$\mathrm{Homeo}_{cpt}(Z)$ contains a net $(g_\beta)_{\beta < \tau}$ of distinct elements such that $\lim g_\beta = \mathrm{id}_Z$ and $\mathrm{supp}\, g_\beta \nsubseteq K_\alpha$ whenever $\alpha < \beta$.*

*Then the coproduct (disjoint union) $Y \amalg Z$ does not have CSHP.*

**Proof.** Since $Y$ is compact, one has $\beta(Y \amalg Z) = Y \amalg \beta Z$. The family $\{Y \cup K_\alpha\}_{\alpha < \tau}$ is cofinal in $(\mathscr{K}(Y \amalg Z), \subseteq)$; in particular, it is directed.

Put $G := \mathrm{Homeo}_{cpt}(Y \amalg Z)$ and $G_\alpha := \mathrm{Homeo}_{Y \cup K_\alpha}(Y \amalg Z)$. We construct a subset $S \subseteq G$ that satisfies the conditions of Lemma 1:

(1)    $S$ is $\tau$-discrete in $G$;

(2)    $|S \cap G_\alpha| < \tau$ for all $\alpha < \tau$; and

(3)    $\mathrm{id}_{Y \amalg Z} \in \overline{S} \backslash S$.

This will show that $G \neq \operatorname*{colim}_{\alpha < \tau} G_\alpha$, and thus $Y \amalg Z$ does not have CSHP.

Let $S' \subseteq \mathrm{Homeo}(Y)$ be a $\tau$-discrete subset such that $|S'| = \tau$ and $S'$ is not closed. Without loss of generality, we may assume that $\mathrm{id}_Y \in \overline{S'} \backslash S'$. Let $S' = \{f_\alpha \mid \alpha < \tau\}$ be an injective enumeration of $S'$. For $\alpha < \tau$, put $h_\alpha := f_\alpha \amalg g_\alpha$. Clearly, $h_\alpha \in G$, because $Y$ and $Z$ are clopen subsets of $Y \amalg Z$.

Put $S := \{h_\alpha \mid \alpha < \tau\}$. We verify that $S$ satisfies properties (1), (2), and (3).

(1) Put $H := \{h \in G \mid h(Y) = Y\}$. Since $Y$ is a compact-open subset of $Y \amalg Z$, the subgroup $H$ is open (and in particular, closed) in $G$, and so it suffices to show that $S$ is $\tau$-discrete in $H$. Let $\iota_Y \colon Y \to Y \amalg Z$ denote the canonical embedding. The composite

$$\mathscr{C}(Y \amalg \beta Z, Y \amalg \beta Z) \xrightarrow{\mathscr{C}(\beta\iota_Y, Y \amalg \beta Z)} \mathscr{C}(Y, Y \amalg \beta Z) \tag{11}$$

is continuous, where the function spaces are equipped with the compact-open topology. Thus, its restriction to $H$ and corestriction to $\mathrm{Homeo}(Y) \subseteq \mathscr{C}(Y, Y \amalg \beta Z)$,

$$\begin{aligned} \Gamma \colon H &\longrightarrow \mathrm{Homeo}(Y) \\ h &\longmapsto h_{|Y} \end{aligned} \tag{12}$$

is a continuous group homomorphism. The restriction $\Gamma_{|S}$ is injective (because $\Gamma(h_\alpha) = f_\alpha$), and $\Gamma(S) = S'$ is $\tau$-discrete in $\mathrm{Homeo}(Y)$. Therefore, by Proposition 2, $S$ is $\tau$-discrete in $H$.

(2) For $\beta < \tau$, $h_\beta \in S \cap G_\alpha$ if and only if $(\mathrm{supp}\, f_\beta) \cup (\mathrm{supp}\, g_\beta) = \mathrm{supp}\, h_\beta \subseteq Y \cup K_\alpha$, or equivalently, $\mathrm{supp}\, g_\beta \subseteq K_\alpha$. By the assumptions on $(g_\beta)_{\beta < \tau}$, the latter is possible only if $\beta \leq \alpha$. Therefore,

$$|S \cap G_\alpha| = |\{\beta \mid \mathrm{supp}\, g_\beta \subseteq K_\alpha\}| \leq |\alpha| < \tau. \tag{13}$$

(3) Since $f_\alpha \neq \mathrm{id}_Y$ for every $\alpha < \tau$, it follows that $h_\alpha \neq \mathrm{id}_{Y \amalg Z}$, and thus $\mathrm{id}_{Y \amalg Z} \notin S$. It remains to show that $\mathrm{id}_{Y \amalg Z} \in \overline{S}$. To that end, let $W$ be an entourage of the diagonal in $(Y \amalg \beta Z)^2$. Then there is an entourage $U$ of the diagonal in $Y \times Y$ and an entourage $V$ of the diagonal in $\beta Z \times \beta Z$ such that $U \cup V \subseteq W$. Since $\lim g_\beta = \mathrm{id}_Z$, there is $\alpha_0 < \tau$ such that for $(\beta g_\alpha(z), z) \in V$ for every $z \in \beta Z$ and $\alpha \geq \alpha_0$. One has

$$|\{f_\alpha \mid \alpha < \alpha_0\}| \leq |\alpha_0| < \tau, \tag{14}$$

and thus $\{f_\alpha \mid \alpha < \alpha_0\}$ is closed, because $S'$ is $\tau$-discrete. Therefore,

$$\mathrm{id}_Y \in \overline{S' \backslash \{f_\alpha \mid \alpha < \alpha_0\}} = \overline{\{f_\alpha \mid \alpha \geq \alpha_0\}}. \tag{15}$$

In particular, there is $\alpha_1 \geq \alpha_0$ such that $(f_{\alpha_1}(y), y) \in U$ for every $y \in Y$. Hence,

$$(\beta h_{\alpha_1}(x), x) = ((f_{\alpha_1} \amalg g_{\alpha_1})(x), x) \in U \cup V \subseteq W \tag{16}$$

for every $x \in Y \amalg \beta Z$, as desired.  $\square$

## 4. Construction of $\tau$-Discrete Subsets

**Theorem D.** *Let $\alpha$ be an infinite limit ordinal with $\tau := \mathrm{cf}(\alpha)$ and put $Y := \alpha + 1$ with the order topology. Then $\mathrm{Homeo}(Y)$ contains a $\tau$-discrete subset of cardinality $\tau$ that is not closed.*

The proof of Theorem D is broken down into several lemmas. First, the special case where the ordinal has countable cofinality is proven. Then, the theorem is reduced to the case where $\alpha = \omega^\beta$ for an infinite limit ordinal $\beta$. (Here, and throughout this paper, $\omega^\beta$ means ordinal exponentiation, not cardinal exponentiation.)

**Proposition 4.** *Let $\alpha$ be an infinite limit ordinal and put $Y := \alpha + 1$ with the order topology. Suppose that $\{f_j\}_{j \in \mathbb{J}} \subseteq \mathrm{Homeo}(Y)$ is a net satisfying that for every $\xi < \alpha$ there is $j_0 \in \mathbb{J}$ such that $f_{j|[0,\xi]} = \mathrm{id}_{[0,\xi]}$ for every $j \geq j_0$. Then $\lim f_j = \mathrm{id}_Y$ in $\mathrm{Homeo}(Y)$.*

**Proof.** Let $U$ be an entourage of the diagonal $\Delta_Y$ in $Y \times Y$. Then $U$ is a neighborhood of the point $(\alpha, \alpha) \in U$, and so there is $\xi < \alpha$ such that $(\xi, \alpha] \times (\xi, \alpha] \subseteq U$. Let $j_0 \in \mathbb{J}$ be such that $f_{j|[0,\xi]} = \mathrm{id}_{[0,\xi]}$ for every $j \geq j_0$. Then, for every $j \geq j_0$ and $x \in Y$,

$$(f_j(x), x) \in \Delta_Y \cup ((\xi, \alpha] \times (\xi, \alpha]) \subseteq U, \tag{17}$$

as desired. $\square$

**Lemma 3.** *Let $\alpha$ be an infinite limit ordinal with countable cofinality and put $Y := \alpha + 1$ with the order topology. Then $\mathrm{Homeo}(Y)$ contains a countable subset that is not closed.*

**Proof.** Let $\{\alpha_n\}_{n < \omega}$ be a strictly increasing cofinal sequence in $\alpha$. Let $f_n \colon Y \to Y$ denote the transposition

$$f_n(x) := \begin{cases} \alpha_n + 2 & x = \alpha_n + 1 \\ \alpha_n + 1 & x = \alpha_n + 2 \\ x & \text{otherwise.} \end{cases} \tag{18}$$

Since $f_n$ is the identity for all but two isolated points, it is a homeomorphism of $Y$. Furthermore, $\lim f_n = \mathrm{id}_Y$ in $\mathrm{Homeo}(Y)$ by Proposition 4, because the $\{\alpha_n\}_{n < \omega}$ are cofinal and increasing. Therefore, $S := \{f_n \mid n < \omega\}$ is a countable subset that is not closed. $\square$

**Lemma 4.** *Let $\beta$ be an infinite limit ordinal with a strictly increasing cofinal family $\{\beta_\delta\}_{\delta < \tau}$, put $\alpha := \omega^\beta$, and put $Y := \alpha + 1$ with the order topology. Then $\mathrm{Homeo}(Y)$ contains a family of non-trivial homeomorphisms $\{f_\delta\}_{\delta < \tau}$ such that*

(a) $f_\delta([\omega^{\beta_\delta+1}, \alpha]) \subseteq [\omega^{\beta_\delta+1}, \alpha]$ *for every $\delta < \tau$, and*
(b) *for every $\gamma < \tau$ and distinct $\delta_1, \delta_2 \leq \gamma$,*

$$f_{\delta_1}(\omega^{\beta_\gamma+1} + 1) \neq f_{\delta_2}(\omega^{\beta_\gamma+1} + 1). \tag{19}$$

**Proof.** Fix an ordinal $\delta < \tau$. Since $\beta_\delta < \beta$, there is an ordinal $\rho_\delta$ such that $\beta = \beta_\delta + 1 + \rho_\delta$. Then $\alpha = \omega^\beta = \omega^{\beta_\delta+1} \omega^{\rho_\delta}$, and every $x < \alpha$ may be uniquely represented in the form

$$x = \omega^{\beta_\delta+1} \varepsilon + \omega^{\beta_\delta} m + \eta, \tag{20}$$

where $\varepsilon < \omega^{\rho_\delta}$, $m < \omega$, and $\eta < \omega^{\beta_\delta}$. Indeed, let

$$x = \omega^{\nu_1} k_1 + \cdots + \omega^{\nu_n} k_n \tag{21}$$

be the Cantor's normal form of $x$, that is, $x \geq \nu_1 > \cdots > \nu_n$ and $k_1, \cdots, k_n$ are non-zero natural numbers ([9], 2.26). Put $\eta := \sum\limits_{\nu_i < \beta_\delta} \omega^{\nu_i} k_i$, where the sums are formed in the same order as in the Cantor's normal form. Clearly, $\eta < \omega^{\beta_\delta}$. For every $i$ such that $\nu_i > \beta_\delta$, there is an ordinal $\mu_i$ such that $\nu_i := \beta_\delta + 1 + \mu_i$. Thus,

$$x = \omega^{\nu_1} k_1 + \cdots + \omega^{\nu_n} k_n$$

$$= \sum_{\nu_i > \beta_\delta} \omega^{\nu_i} k_i + \omega^{\beta_\delta} m + \underbrace{\sum_{\nu_i < \beta_\delta} \omega^{\nu_i} k_i}_{\eta}$$

$$= \sum_{\nu_i > \beta_\delta} \omega^{\beta_\delta + 1 + \mu_i} k_i + \omega^{\beta_\delta} m + \eta$$

$$= \omega^{\beta_\delta + 1} \underbrace{\sum_{\nu_i > \beta_\delta} \omega^{\mu_i} k_i}_{\varepsilon} + \omega^{\beta_\delta} m + \eta$$

$$= \omega^{\beta_\delta + 1} \varepsilon + \omega^{\beta_\delta} m + \eta, \tag{22}$$

where $m = 0$ if $\nu_i \neq \beta_\delta$ for any $i$ and $m = k_{i_0}$ if $\nu_{i_0} = \beta_\delta$. (The uniqueness of this representation follows from the uniqueness of the Cantor's normal form.)

We construct now $f_\delta$. Let $\varphi_\delta \colon \omega \to \omega$ be a bijection such that $\varphi_\delta(0) \neq 0$. Put

$$f_\delta(\omega^{\beta_\delta + 1} \varepsilon + \omega^{\beta_\delta} m + \eta) := \begin{cases} \omega^{\beta_\delta + 1} \varepsilon + \omega^{\beta_\delta} \varphi_\delta(m) + \eta & \eta > 0 \\ \omega^{\beta_\delta + 1} \varepsilon + \omega^{\beta_\delta} (\varphi_\delta(m-1) + 1) & \eta = 0, m > 0 \\ \omega^{\beta_\delta + 1} \varepsilon & \eta = m = 0 \\ \alpha & x = \alpha. \end{cases} \tag{23}$$

It is clear that $f_\delta \neq \mathrm{id}_Y$. We show that $f_\delta \in \mathrm{Homeo}(Y)$. First, we note that $f_\delta$ is a bijection, whose inverse is of the same form with $\varphi_\delta$ replaced with $\varphi_\delta^{-1}$. Thus, it suffices to show that $f_\delta$ is continuous. Let $x \in Y$ be an infinite limit ordinal, and let $\{x_j\} \subseteq Y$ be a net converging to $x$. Without loss of generality, we may assume that $x_j < x$ for every $j$ and that the $\{x_j\}$ are non-decreasing. We distinguish the cases used to define $f_\delta$.

*Case 1.* If $x = \omega^{\beta_\delta + 1} \varepsilon + \omega^{\beta_\delta} m + \eta$, where $\eta > 0$, then without loss of generality, we may assume that $x_j > \omega^{\beta_\delta + 1} \varepsilon + \omega^{\beta_\delta} m$, and thus $x_j = \omega^{\beta_\delta + 1} \varepsilon + \omega^{\beta_\delta} m + \eta_j$, where $0 < \eta_j$ and $\{\eta_j\}$ converges to $\eta$. Therefore,

$$f_\delta(x_j) = \omega^{\beta_\delta + 1} \varepsilon + \omega^{\beta_\delta} \varphi_\delta(m) + \eta_j \longrightarrow \omega^{\beta_\delta + 1} \varepsilon + \omega^{\beta_\delta} \varphi_\delta(m) + \eta = f_\delta(x). \tag{24}$$

*Case 2.* If $x = \omega^{\beta_\delta + 1} \varepsilon + \omega^{\beta_\delta} m$, where $m > 0$, then without loss of generality, we may assume that $x_j > \omega^{\beta_\delta + 1} \varepsilon + \omega^{\beta_\delta} (m - 1)$, and thus $x_j = \omega^{\beta_\delta + 1} \varepsilon + \omega^{\beta_\delta} (m - 1) + \eta_j$, where $0 < \eta_j$ and $\{\eta_j\}$ converges to $\omega^{\beta_\delta}$. Therefore,

$$f_\delta(x_j) = \omega^{\beta_\delta + 1} \varepsilon + \omega^{\beta_\delta} \varphi_\delta(m-1) + \eta_j \longrightarrow \omega^{\beta_\delta + 1} \varepsilon + \omega^{\beta_\delta} (\varphi_\delta(m-1)) + \omega^{\beta_\delta} = f_\delta(x). \tag{25}$$

*Cases 3 and 4.* If $x = \omega^{\beta_\delta + 1} \varepsilon$ where $\varepsilon \leq \omega^{\rho_\delta}$, then $x_j = \omega^{\beta_\delta + 1} \varepsilon_j + \omega^{\beta_\delta} m_j + \eta_j$ where $\varepsilon_j < \varepsilon$. Since $\{x_j\}$ are non-decreasing, the $\{\varepsilon_j\}$ are non-decreasing. Put $\varepsilon_0 := \sup \varepsilon_j$. Clearly, $\varepsilon_0 \leq \varepsilon$. If $\varepsilon_0 = \varepsilon$, then $\omega^{\beta_\delta + 1} \varepsilon_j \longrightarrow \omega^{\beta_\delta + 1} \varepsilon = x$, and $\omega^{\beta_\delta + 1} \varepsilon_j \leq f_\delta(x_j) \leq \omega^{\beta_\delta + 1} \varepsilon = x$; hence, $f_\delta(x_j) \longrightarrow x = f_\delta(x)$. If $\varepsilon_0 < \varepsilon$, then

$$\omega^{\beta_\delta + 1} (\varepsilon_0 + 1) \geq \omega^{\beta_\delta + 1} (\varepsilon_j + 1) \geq x_j \longrightarrow \omega^{\beta_\delta + 1} \varepsilon, \tag{26}$$

and thus $\varepsilon_0 + 1 = \varepsilon$. Since $x_j > \omega^{\beta_\delta + 1} \varepsilon_0$ eventually, without loss of generality, we may assume that $\varepsilon_j = \varepsilon_0$ for all $j$, and $0 < m_j$ and $m_j \to \omega$. Since $\varphi_\delta$ is a bijection, it follows that $\varphi_\delta(m_j) \longrightarrow \omega$ and $\varphi_\delta(m_j - 1) \longrightarrow \omega$. Therefore,

$$f(x_j) \geq \omega^{\beta_\delta + 1} \varepsilon_0 + \omega^{\beta_\delta} \min\{\varphi_\delta(m_j - 1), \varphi_\delta(m_j)\} \longrightarrow \omega^{\beta_\delta + 1} \varepsilon_0 + \omega^{\beta_\delta} \omega = \omega^{\beta_\delta + 1} \varepsilon = f_\delta(x). \tag{27}$$

This shows that $f_\delta$ and $f_\delta^{-1}$ are continuous, and therefore $f_\delta \in \mathrm{Homeo}(Y)$.

Property (a) follows directly from the definition and the more general property of $f_\delta$ that for every $\varepsilon \leq \omega^{\rho_\delta}$, if $x \geq \omega^{\beta_\delta + 1} \varepsilon$, then $f_\delta(x) \geq \omega^{\beta_\delta + 1} \varepsilon$.

Lastly, we prove property (b). Let $\gamma < \tau$ and $\delta_1, \delta_2 \leq \gamma$ be distinct. There are ordinals $\zeta_i$ such that $\beta_\gamma + 1 = \beta_{\delta_i} + 1 + \zeta_i$ (for $i = 1, 2$). By the definition,

$$f_{\delta_i}(\omega^{\beta_\gamma+1} + 1) = f_{\delta_i}(\omega^{\beta_{\delta_i}+1+\zeta_i} + 1) = f_{\delta_i}(\omega^{\beta_{\delta_i}+1}\omega^{\zeta_i} + 1) \tag{28}$$

$$= \omega^{\beta_{\delta_i}+1}\omega^{\zeta_i} + \omega^{\beta_{\delta_i}+1}\varphi_{\delta_i}(0) + 1 = \omega^{\beta_\gamma+1} + \omega^{\beta_{\delta_i}+1}\varphi_{\delta_i}(0) + 1. \tag{29}$$

Since $\delta_1 \neq \delta_2$ and $\varphi_{\delta_i}(0) \neq 0$, it follows that $\omega^{\beta_{\delta_1}+1}\varphi_{\delta_1}(0) \neq \omega^{\beta_{\delta_2}+1}\varphi_{\delta_2}(0)$. Therefore, one obtains $f_{\delta_1}(\omega^{\beta_\gamma+1} + 1) \neq f_{\delta_2}(\omega^{\beta_\gamma+1} + 1)$, as desired. $\square$

**Lemma 5.** *Let $\beta$ be an infinite limit ordinal with a strictly increasing cofinal family $\{\beta_\delta\}_{\delta<\tau}$, put $\alpha := \omega^\beta$, and put $Y := \alpha + 1$ with the order topology. Let $\psi_\delta \colon [0, \alpha] \to [\omega^{\beta_\delta} + 1, \alpha]$ denote the homeomorphism defined by $\psi_\delta(x) := \omega^{\beta_\delta} + 1 + x$. For $f \in \mathrm{Homeo}(Y)$, define $\Psi_\delta(f)$ by*

$$\Psi_\delta(f) \colon [0, \alpha] \longrightarrow [0, \alpha]$$

$$x \longmapsto \begin{cases} x & x \in [0, \omega^{\beta_\delta}] \\ \psi_\delta f \psi_\delta^{-1}(x) & x > \omega^{\beta_\delta}. \end{cases} \tag{30}$$

(a)   $\Psi_\delta(f) \in \mathrm{Homeo}(Y)$ *for every $f \in \mathrm{Homeo}(Y)$ and $\delta < \tau$.*

(b)   *If $f \in \mathrm{Homeo}(Y)$ and $\delta < \tau$ are such that $f([\omega^{\beta_\delta+1}, \alpha]) \subseteq [\omega^{\beta_\delta+1}, \alpha]$, then $\Psi_\delta(f)(x) = f(x)$ for every $x \in [\omega^{\beta_\delta+1}, \alpha]$.*

(c)   *For every family $\{f_\delta\}_{\delta<\tau} \subseteq \mathrm{Homeo}(Y)$, the net $\{\Psi_\delta(f_\delta)\}_{\delta<\tau}$ converges to $\mathrm{id}_Y$ in $\mathrm{Homeo}(Y)$.*

**Proof.** (a) Let $f \in \mathrm{Homeo}(Y)$. Since $\Psi(f)$ is continuous on the clopen sets $[0, \omega^{\beta_\delta} + 1)$ and $(\omega^{\beta_\delta}, \alpha]$, it is continuous on $Y$. Furthermore, it is easily seen that $\Psi(f)^{-1} = \Psi(f^{-1})$, and thus $\Psi_\delta(f) \in \mathrm{Homeo}(Y)$.

(b) Let $x \in [\omega^{\beta_\delta+1}, \alpha]$. Then $x = \omega^{\beta_\delta+1} + y$ for some $y \in [0, \alpha]$, and so

$$\psi_\delta(x) = \psi_\delta(\omega^{\beta_\delta+1} + y) = \omega^{\beta_\delta} + 1 + \omega^{\beta_\delta+1} + y = \omega^{\beta_\delta+1} + y = x. \tag{31}$$

Thus, $\psi_\delta^{-1}(x) = x$, and $f\psi_\delta^{-1}(x) = f(x)$. Therefore, $\psi_\delta f \psi_\delta^{-1}(x) = f(x)$, because $f(x) \in [\omega^{\beta_\delta+1}, \alpha]$ by our assumption.

(c) Since $\{\omega^{\beta_\delta}\}_{\delta<\tau}$ is a strictly increasing cofinal family in $\alpha$ and $\Psi_\delta(f_\delta)$ is the identity on $[0, \omega^{\beta_\delta}]$, it follows by Proposition 4 that $\lim \Psi_\delta(f_\delta) = \mathrm{id}_Y$. $\square$

**Proof of Theorem D.** By Lemma 3, we may assume that $\tau > \omega$. Let

$$\alpha = \omega^{\beta_1}k_1 + \cdots + \omega^{\beta_n}k_n \tag{32}$$

be the Cantor's normal form of $\alpha$, that is, $\alpha \geq \beta_1 > \cdots > \beta_n$ and $k_1, \cdots, k_n$ are non-zero natural numbers ([9], 2.26). Since $\tau > \omega$, it follows that $\mathrm{cf}(\beta_n) \neq 1$, and so

$$\mathrm{cf}(\beta_n) = \mathrm{cf}(\omega^{\beta_n}) = \tau.$$

The space $\omega^{\beta_n} + 1$ embeds as a clopen subset into $Y$, and so $\mathrm{Homeo}(\omega^{\beta_n} + 1)$ embeds as a closed subgroup into $\mathrm{Homeo}(Y)$. Therefore, without loss of generality, we may assume that $\alpha := \omega^\beta$, where $\beta$ is an ordinal of uncountable cofinality.

Let $\{\beta_\delta\}_{\delta<\tau}$ be a strictly increasing cofinal family in $\beta$, and let $\{f_\delta\}_{\delta<\tau}$ be a family in $\mathrm{Homeo}(Y)$ as provided by Lemma 4. Put $h_\delta := \Psi_\delta(f_\delta)$, where $\Psi_\delta$ is as in Lemma 5, and set $S := \{h_\delta \mid \delta < \tau\}$. By Lemma 4(a), $f_\delta([\omega^{\beta_\delta+1}, \alpha]) \subseteq [\omega^{\beta_\delta+1}, \alpha]$ for every $\delta < \tau$, and thus by Lemma 5(b),

$$h_\delta(x) = \Psi_\delta(f_\delta)(x) = f_\delta(x) \text{ for every } x \in [\omega^{\beta_\delta+1}, \alpha]. \tag{33}$$

We show that $S$ is $\tau$-discrete but not closed. (Since $|S| \leq \tau$ by the construction, $|S| = \tau$ follows from these two.) By Lemma 5(c), $h_\delta = \Psi_\delta(f_\delta)$ converges to $\mathrm{id}_Y$. Since $f_\delta \neq \mathrm{id}_Y$, it follows that $\Psi_\delta(f_\delta) \neq \mathrm{id}_Y$. Thus, $S$ is not closed.

Let $C \subseteq \tau$ be a subset such that $|C| < \tau$. Put $\gamma := \sup C$. Since $\tau$ itself is a regular cardinal, $\gamma < \tau$. Suppose that $\{h_{\delta_j}\}_{j \in \mathbb{J}}$ is a net in $\{h_\delta \mid \delta \in C\}$ that converges to $h \in \mathrm{Homeo}(Y)$. Then, in particular, $\lim h_{\delta_j}(\omega^{\beta_\gamma+1} + 1) = h(\omega^{\beta_\gamma+1} + 1)$. Thus, by (33),

$$\lim h_{\delta_j}(\omega^{\beta_\gamma+1} + 1) = \lim f_{\delta_j}(\omega^{\beta_\gamma+1} + 1) = h(\omega^{\beta_\gamma+1} + 1). \tag{34}$$

Since $\omega^{\beta_\gamma+1} + 1$ is an isolated point, so is its homeomorphic image $h(\omega^{\beta_\gamma+1} + 1)$. Therefore, the net is eventually constant, and so there is $j_0 \in \mathbb{J}$ such that

$$f_{\delta_j}(\omega^{\beta_\gamma+1} + 1) = h_{\delta_{j_0}}(\omega^{\beta_\gamma+1} + 1) \text{ for } j \geq j_0.$$

Hence, by Lemma 4(b), $\delta_j = \delta_{j_0}$ for every $j \geq j_0$. It follows that $h = \lim h_{\delta_j} = h_{\delta_{j_0}} \in S$, as desired. $\square$

**Corollary 1.** *Suppose that $\lambda = \alpha + \xi$, where $\alpha$ is an ordinal, $\xi > 0$ is an infinite limit ordinal, and $\alpha \geq \mathrm{cf}(\xi)$. Then $X = \lambda$ does not have CSHP.*

**Proof.** Put $\tau := \mathrm{cf}(\xi)$. Without loss of generality, we may assume that $\tau > \omega$. (If $\tau = \omega$, then $\xi$ contains an increasing cofinal sequence $\{\xi_n\}_{n<\omega}$, and thus

$$D := \{\alpha + \xi_n + 1 \mid n < \omega\}$$

is an infinite discrete clopen subset of $X$, and so by Lemma 2(b), $X$ does not have CSHP.)

Since $\xi$ is an infinite ordinal, $1 + \xi = \xi$, and so $\lambda = (\alpha + 1) + \xi$. Thus, $X \cong (\alpha + 1) \amalg \xi$, and $(\tau + 1) \amalg \xi$ embeds into $X$ as a clopen subset. Therefore, by Lemma 2(a), it suffices to show that $(\tau + 1) \amalg \xi$ does not have CSHP.

Put $Y := \tau + 1$ and $Z := \xi$. We verify that the conditions of Theorem C are satisfied. Clearly, $Y$ is compact Hausdorff and $Z$ is locally compact Hausdorff. Let $\{\xi_\gamma \mid \gamma < \tau\}$ be cofinal and increasing in $\xi$. Put $K_\gamma := [0, \xi_\gamma]$ for $\gamma < \tau$. Then $\{K_\gamma\}_{\gamma < \tau}$ is cofinal in $\mathscr{K}(Z)$.

(I) By Theorem D, $\mathrm{Homeo}(Y)$ contains a $\tau$-discrete subset of cardinality $\tau$ that is not closed.

(II) For $\gamma < \tau$, let $f_\gamma$ denote the transposition that interchanges $\xi_\gamma + 1$ and $\xi_\gamma + 2$, and leave every other point fixed. Clearly, $f_\gamma \in \mathrm{Homeo}_{cpt}(Z)$ and $\mathrm{supp}(f_\delta) \cap K_\gamma = \varnothing$ for every $\gamma < \delta < \tau$. Since $\tau > \omega$, one has $\beta Z = \xi + 1$ (cf. [10], 5N1). For every $\delta < \xi$, there is $\gamma_0 < \tau$ such that $\xi_\gamma > \delta$ for every $\gamma \geq \gamma_0$, and so $f_{\gamma|[0,\delta]} = \mathrm{id}_{[0,\delta]}$. Therefore, by Proposition 4, $\lim f_\gamma = \mathrm{id}_Z$. $\square$

## 5. Products of Ordinals

In this section, we prove Theorem A, which provides necessary and sufficient conditions for a product of ordinals to have CSHP.

**Theorem A.** *Let $X := \lambda_1 \times \cdots \lambda_k \times \mu_1 \times \cdots \times \mu_l$ equipped with the product topology, where $\lambda_1, \ldots, \lambda_k$ are infinite limit ordinals and $\mu_1, \ldots, \mu_k$ are successor ordinals. The space $X$ has CSHP if and only if there is an uncountable regular cardinal $\kappa$ such that $\lambda_1 = \cdots = \lambda_k = \kappa$ and $\mu_i \leq \kappa$ for every $i = 1, \ldots, l$.*

Sufficiency was proven in the authors' previous work ([5], Theorem D(c)), and so it is only necessity that has to be shown. We first prove a special case of Theorem A.

**Theorem 1.** *Let $\lambda$ be an infinite limit ordinal, and let $X$ be $\lambda$ equipped with the order topology. If the space $X$ has CSHP, then $\lambda$ is an uncountable regular cardinal.*

**Proof.** If $\lambda$ had countable cofinality, then it would contain an infinite discrete clopen subset. It would follow then by Lemma 2(b), that $\lambda$ does not have CSHP, contrary to our assumption. Thus, $\mathrm{cf}(\lambda) > \omega$.

Next, we show that $\lambda = \omega^\beta$ for some ordinal $\beta$. Let

$$\lambda = \omega^{\beta_1} k_1 + \cdots + \omega^{\beta_n} k_n \tag{35}$$

be the Cantor's normal form of $\lambda$, that is, $\lambda \geq \beta_1 > \cdots > \beta_n$ and $k_1, \cdots, k_n$ are non-zero natural numbers ([9], 2.26). Put $\alpha := \omega^{\beta_1} k_1 + \cdots + \omega^{\beta_n}(k_n - 1)$ and $\xi = \omega^{\beta_n}$. It follows from $\beta_1 > \cdots > \beta_n$ that either $\alpha \geq \xi$ or $\alpha = 0$. Since $\lambda$ is an infinite limit ordinal, one has $\beta_n > 0$, and so $\xi > 0$ and $\xi$ is a limit ordinal. Thus, by Corollary 1 applied to $\lambda = \alpha + \xi$, one obtains that $\alpha < \mathrm{cf}(\xi) \leq \xi$. Therefore, $\alpha = 0$. In other words, $n = 1$ and $k_1 = 1$, and $\lambda = \omega^{\beta_1}$.

We show that $\lambda$ is a regular cardinal by proving that $\mathrm{cf}(\lambda) \geq \lambda$. Put $\tau := \mathrm{cf}(\lambda)$. One has $\tau = \omega^\tau$ (ordinal exponentiation), because $\tau$ is an uncountable cardinal, and for every ordinal $\theta < \tau$, one has $|\omega^\theta| = \max(\omega, |\theta|) < \tau$ (where $\omega^\theta$ is ordinal exponentiation).

If $\tau < \beta_1$, then $\omega^\tau < \omega^{\beta_1}$, and so

$$\tau + \lambda = \omega^\tau + \omega^{\beta_1} = \omega^{\beta_1} = \lambda, \tag{36}$$

([11], Theorem 1 from XIV.6 and Theorem 1 from XIV.19). Thus, by Corollary 1, $\tau + \lambda = \lambda$ cannot have CSHP, contrary to our assumption. Hence, $\tau \geq \beta_1$, and

$$\tau = \omega^\tau \geq \omega^{\beta_1} = \lambda, \tag{37}$$

as desired.   $\square$

We proceed now to prove Theorem A.

**Proof of Theorem A.** Suppose that $X$ has CSHP. Without loss of generality, we may assume that $k \geq 1$. Put $\kappa := \min \lambda_i$. Since $\kappa$ embeds as a clopen subset of $X$, by Lemma 2(a), $\kappa$ has CSHP. Thus, by Theorem 1, $\kappa$ is an uncountable regular cardinal.

Put $Y := \kappa + 1$ and $Z := \kappa$ with the order topology. By Theorem D, $\mathrm{Homeo}(Y)$ contains a $\kappa$-discrete subset of cardinality $\kappa$ that is not closed. The space $Z$ is zero-dimensional, locally compact, pseudocompact, and $\kappa := \mathrm{cf}(\mathscr{K}(Z), \subseteq)$. Therefore, by Theorem B, the product $Y \times Z$ does not have CSHP.

Assume that there is an $i$ such that $\lambda_i > \kappa$ or $\mu_i > \kappa$. Then $\kappa + 1 \leq \lambda_i$ or $\kappa + 1 \leq \mu_i$, respectively, and thus, by Lemma 2(a), $(\kappa + 1) \times \kappa$ has CSHP, being homeomorphic to a clopen subset of $X$. This contradiction shows that all $\lambda_i$ are equal and $\mu_i \leq \kappa$ for every $i$.   $\square$

**Corollary 2.** *Let $\alpha \leq \beta$ be infinite ordinals. The disjoint union (coproduct) $\alpha \amalg \beta$ has CSHP if and only if one of the following conditions hold:*

*(a)*   *$\alpha$ and $\beta$ are successor ordinals, or;*
*(b)*   *$\beta$ is an uncountable regular cardinal, and in addition, $\alpha = \beta$ or $\alpha$ is a successor ordinal.*

**Proof.** Suppose that $\alpha \amalg \beta$ has CSHP. By Lemma 2(a), $\alpha$ and $\beta$ both have CSHP. Thus, by Theorem 1, $\alpha$ and $\beta$ are either successor ordinals or uncountable regular cardinals. If $\alpha = \beta$, then we are done. On the other hand, if $\alpha < \beta$, then $\alpha + 1$ is a clopen subset of $\beta$, and thus, by Lemma 2(a), $\alpha \amalg (\alpha + 1) \cong (\alpha + 1) + \alpha = \alpha + \alpha$ has CSHP, being a clopen subset of $\alpha \amalg \beta$. By Theorem 1, $\alpha$ is a successor ordinal, because $\alpha + \alpha$ is not a cardinal.

Conversely, if $\alpha$ and $\beta$ are successor ordinals, then $\alpha \amalg \beta$ is compact, and we are done. So, we confine our attention to case (b). Suppose that $\beta$ is an uncountable regular cardinal. If $\alpha = \beta$, then $\alpha \amalg \beta \cong \beta \times 2$ has CSHP by Theorem A. If $\alpha < \beta$ and $\alpha$ is a successor ordinal, then $\alpha \amalg \beta \cong \alpha + \beta = \beta$, which has CSHP by Theorem 1.   $\square$

**Author Contributions:** R.D. and G.L. have read and agreed to the published version of the manuscript.

**Funding:** This research received no external funding.

**Acknowledgments:** We are grateful to Karen Kipper for her kind help in proofreading this paper for grammar and punctuation.

**Conflicts of Interest:** The authors declare no conflict of interest.

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
