# Peer review of "Long Colimits of Topological Groups III: Homeomorphisms of Products and Coproducts"

_axioms, doi:10.3390/axioms10030155_

Round 1

Reviewer 1 Report

The names of the journals in reference 1 and 2 should be modified to abbreviated form. 

Author Response

Thank you for your comment. References 1 and 2 have been corrected  to use abbreviated journal name.

Reviewer 2 Report

Fine article! Minor changes are required.

Author Response

Thank you for the kind words and great suggestions. We have implemented them all:

Preliminaries, line 9: added "of X"  (as suggested by reviewer 2).

Page 4, Theorem B: Added "Hausdorff" (to match formulation on page 2).

Theorem B, pages 2 and 4: Added "that is not compact" to ensure that $\tau$ is infinite (to address the suggestion of reviewer 2).

Preliminaries (p. 3), line 2: Replaced with "is a directed system with respect to inclusion" for greater clarity (as suggested by reviewer 2).

Proposition 2.2 (p. 3), line 2: Added "such" (as suggested by reviewer 2).

Reviewer 3 Report

The paper appears fine, offering an interesting perspective on topological groups. The authors name this paper as "III" in a sequence, and they also refer to another paper they have published as number "I". So the reader is left with the query about what happened to paper "II". May be this is something the authors need to consider before publication.

Author Response

Thank you for your helpful comment. The authors' paper "II" is now mentioned on page 2, line 2, and has been added to the References.